# Indirect Analysis of Concrete Slump Using Different Metaheuristic-Empowered Neural Processors

Hamed Safayenikoo [1,*], Fatemeh Nejati [2] and Moncef L. Nehdi [3,*]

1. Department of Civil Engineering, Chabahar Maritime University, Chabahar 9971778631, Iran
2. Department of Art and Architecture, Faculty of Architecture, Khatam University, Tehran 1991633357, Iran
3. Department of Civil Engineering, McMaster University, Hamilton, ON L8S 4M6, Canada
* Correspondence: h.safayenikoo@cmu.ac.ir (H.S.); nehdim@mcmaster.ca (M.L.N.);
Tel.: +905-525-9140 (ext. 23824) (M.L.N.)

**Abstract:** Estimating the mechanical parameters of concrete is significant towards achieving an efficient mixture design. This research deals with concrete slump analysis using novel integrated models. To this end, four wise metaheuristic techniques of biogeography-based optimization (BBO), salp swarm algorithm (SSA), moth-flame optimization (MFO), and wind driven optimization (WDO) are employed to optimize a popular member of the neural computing family, namely multilayer perceptron (MLP). Four predictive ensembles are constructed to analyze the relationship between concrete slump and seven concrete ingredients including cement, water, slag, fly ash, fine aggregate, superplasticizer, and coarse aggregate. After discovering the optimal complexities by sensitivity analysis, the results demonstrated that the combination of metaheuristic algorithms and neural methods can properly handle the early prediction of concrete slump. Moreover, referring to the calculated ranking scores (RSs), the BBO-MLP (RS = 21) came up as the most accurate model, followed by the MFO-MLP (RS = 17), SSA-MLP (RS = 12), and WDO-MLP (RS = 10). Lastly, the suggested models can be promising substitutes to traditional approaches in approximating the concrete slump.

**Keywords:** sustainable construction; concrete mixture; slump; neural computing; metaheuristic optimization

## 1. Introduction

Having a reliable assessment of concrete slump is a significant task as it is highly linked to the workability of the mixture [1]. However, like many engineering issues, problems such as high complexity and non-linearity are serious obstacles. in the way. Up to now, many experts have tried to analyze slump using traditional and conventional methods. Among those are empirical methods that are developed based on experimental results. Despite the extensive use of such models for analyzing a particular characteristic of concrete, they usually fail in the case of high dimensional problems [2].

In a broader sense, civil engineering has extensively benefitted from recent advances leading to development of new methodologies and technologies for solving issues related to various domains e.g., hydraulic [3], traffic [4], safety [5] and geotechnical [6,7] engineering. Going beyond traditional methods, new approaches are able to deal with dynamic and non-linear conditions [8,9]. For instance, analyzing the strength of various materials (e.g., asphalt [10,11], sand, etc. [12–15]) is among the notable applications.

More specific to structural engineering, complicated types of apparatus have been invented and used to meticulously monitor structural components [16,17] under different conditions (e.g., seismic loading [18–20]). Structural health monitoring is a branch of this field, aimed at damage detection and monitoring failure mechanisms [21–23]. Proper evaluation of structural behaviors entails knowing the response of materials. As for concrete, it can be made with various materials, combined with various ratios depending on the purpose of use [24–27]. In many efforts, engineers have achieved potential methodologies

for analyzing the strength of concrete-based components in structures (e.g., beam [28] and column [29,30]) subjected to complex loading conditions [31,32].

Although developing simulation packages (e.g., finite element model [33], image processing [34–36], etc.) employ reliable methods for such assessments, computer-aided algorithms have resulted in unsupervised and semi-supervised learning approaches that increase the efficiency of traditional ones [37–39]. In this sense, machine learning and deep learning [40,41] have promisingly served in exploring the interrelated relationships between a set of parameters.

According to earlier studies, machine learning models provide a potential solution for approximating the mechanical parameters of concrete [42,43]. Artificial neural networks (ANNs) are non-linear processors which have been extensively used for analyzing the compressive strength (CS) for diverse types of concrete [44–46]. ANNs have also shown reliable performances in dealing with elastic modulus, durability, and corrosion simulation of this material [47–49]. More particularly, such tools have been widely employed in order to estimate the slump of concrete mixtures [50–52]. Öztaş, et al. [53] successfully employed an ANN for estimating slump (as well as the CS) of high strength concrete. Jain, et al. [54] applied an ANN to the results of laboratory tests to predict slump. Nehdi, et al. [55] showed the capability of this intelligent model for predicting concrete slump, filling capacity and segregation. Dias and Pooliyadda [56] used an ANN to estimate the CS and slump of concrete with chemical and/or mineral admixtures. They also showed the superiority of the ANN to multiple regression models in this task.

More recently, highly non-linear and complicated engineering problems have driven scientists to employ metaheuristic techniques for diverse optimization objectives [57–61]. Focusing on the civil engineering domain, their application covers a broad range from structural health monitoring [62] to material strength modeling [63]. In these algorithms, an optimal solution is sought by a set of populations that update their situation within the defined search space. Thus, they can properly assist intelligent models to attain a system preserved from prevalent training issues [64–66].

In the case of concrete parameter modeling, a metaheuristic algorithm tries to establish the best contribution between the mixture gradients and the purposed parameter. For instance, Bui, et al. [67] applied whale optimization algorithm to an ANN for predicting CS. This algorithm achieved root mean square error (RMSE) of 2.6985 which was comparably lower than the RMSE of dragonfly algorithm (3.3325) and ant colony optimization (3.4452). In another work by Bui, et al. [68], modified firefly algorithm (MFA) was assigned to the similar task for high performance concrete. They compared this model with those in several previous studies (e.g., genetic programming) and found that the MFA-ANN is superior to them. Zhao, et al. [69] proved the proficiency of equilibrium optimizer for analyzing the tensile strength. The hybrid model could improve the performance of ANN (e.g., correlation rose from 0.89 to 0.92).

As for concrete slump, Moayedi, et al. [70] optimized the ANN using ant lion optimization (ALO) in predicting slump. Based on the calculated RMSE and mean absolute error (MAE) of 3.0286 and 3.7788, they found the ALO a promising optimizer for this aim. Moreover, their proposed model surpassed two popular techniques, namely grasshopper optimization algorithm and biogeography-based optimization. Likewise, Chandwani, et al. [2] used genetic algorithm to improve the prediction capability of the ANN. Foong, et al. [71] tested the performance of three potential techniques: electromagnetic field optimization (EFO), teaching-learning-based optimization (TLBO), water cycle algorithm (WCA), and multi-tracker optimization algorithm (MTOA). It was shown that EFO enjoys the highest efficacy.

From the above literature, it can be observed that although regular machine learning methods and genetic-based approaches [72,73] have been sufficiently used for simulating slump, the application of metaheuristic optimizers has not been sufficiently studied in this field. Therefore, this study has been conducted to evaluate the efficiency of biogeography-based optimization (BBO), salp swarm algorithm (SSA), moth-flame optimization (MFO),

and wind driven optimization (WDO) in slump prediction. The principal contribution of the applied algorithms to the above-mentioned problem is characterized by exploring the slump–ingredients relationship through neural computing rules.

## 2. Data and Modeling Methodology

### 2.1. Data

As is known, it is very important to feed machine learning models with proper data. Each dataset must contain two major groups: (a) a number of input variables as independent factors of the problem that influence (b) one or more output variable (s) as a dependent factor. Due to the main objective of the present study, concrete slump is selected as the dependent factor influenced by concrete contents including cement (X1), slag (X2), water (X3), fly ash (X4), superplasticizer (X5), fine aggregate (X6), and coarse aggregate (X7). It should be noted that the mentioned data are provided by Yeh [74] based on the standards of the American Society of Testing and Materials (ASTM) for making concrete specimens as well as the conventional slump test (ASTM C143/C143M-00) [75] for determining the fresh concrete consistence.

The slump values vary from 0.0 to 29.0 cm, and X1, X2, . . . , X7 range in (137.0, 374.0), (0.0, 260.0), (160.0, 240.0), (0.0, 193.0), (4.4, 19.0), (640.6, 902.0), and (708.0, 1049.9), respectively. The histogram of the input factors as well as the concrete slump are depicted in Figure 1. In this study, 80% of data (i.e., 82 samples) are used to train the intelligent models and the rest of the data which are 20% (i.e., 21 samples) are set aside to evaluate the prediction capability of the models.

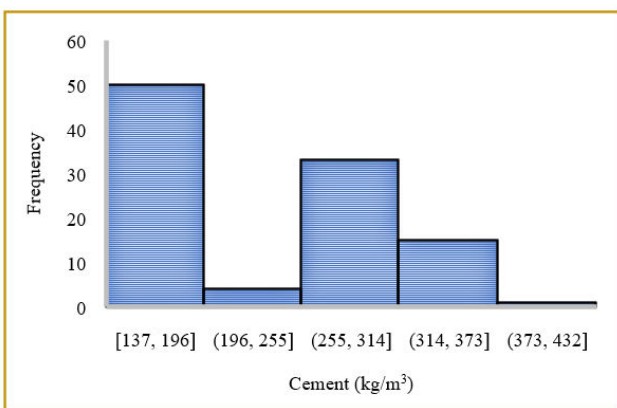

**(a)**

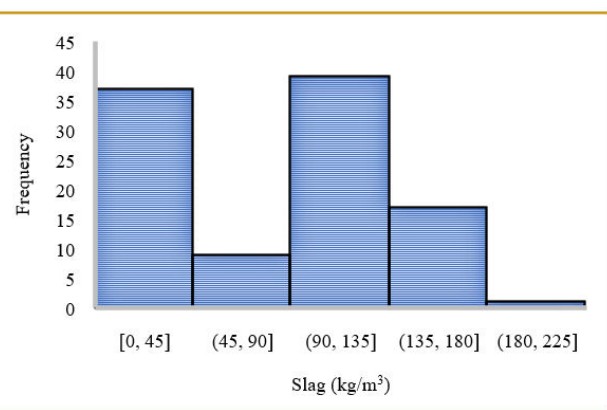

**(b)**

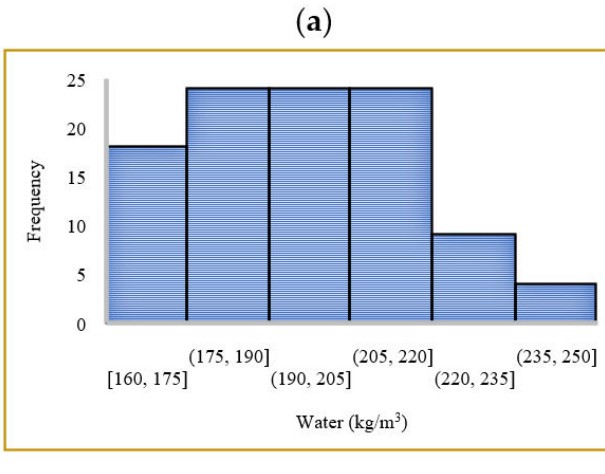

**(c)**

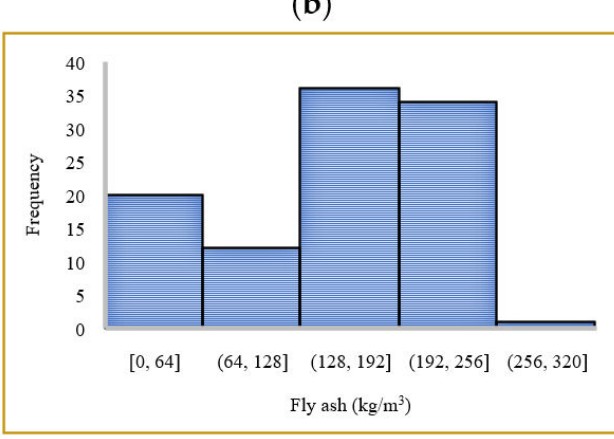

**(d)**

**Figure 1.** *Cont.*

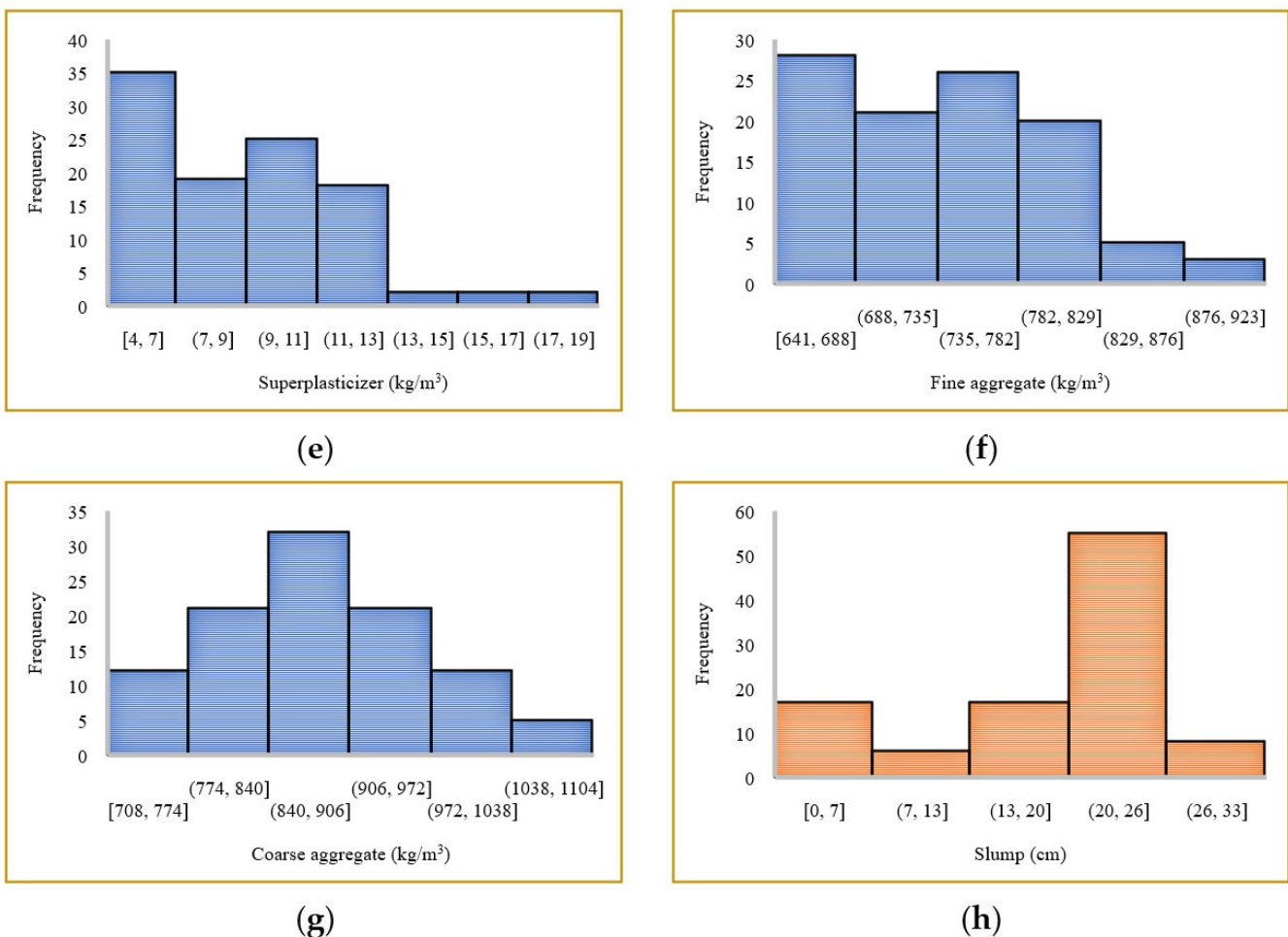

**Figure 1.** The graphical description of (**a**) X1, (**b**) X2, (**c**) X3, (**d**) X4, (**e**) X5, (**f**) X6, (**g**) X7, and (**h**) Slump.

### 2.2. Methodology

### 2.2.1. Artificial Neural Network

The advent of intelligent techniques, like ANNs and ANFIS, has highly influenced the computation of non-linear problems. These methods are known as "soft computing" which are able to analyze the complex relationships between a set of dependent-independent parameters. Based on the idea presented by McCulloch and Pitts [76], ANNs are designed to mimic the way that biological systems work. Among the different types of ANNs (e.g., radial basis function [77] and general regression [78]), multi-layer perceptron (MLP) [79] has been widely used in many studies [80–84]. As the name denotes, an MLP is a multi-layered network where there are a number of processors (called nodes or neurons) in each layer (see Figure 2).

The MLP draws on two main bases, namely learning method backpropagation (BP) [85] and Levenberg–Marquardt [86] training algorithm. Focusing on the MLP mechanism, as Figure 2 outlines, the input of each node ($T$) is multiplied by a weight factor ($W$). Next, a bias factor ($b$) is added to their summation, and finally, the respace is released by applying an activation function ($F$). This function is selected to be a Sigmoid one in the present study (see Figure 3).

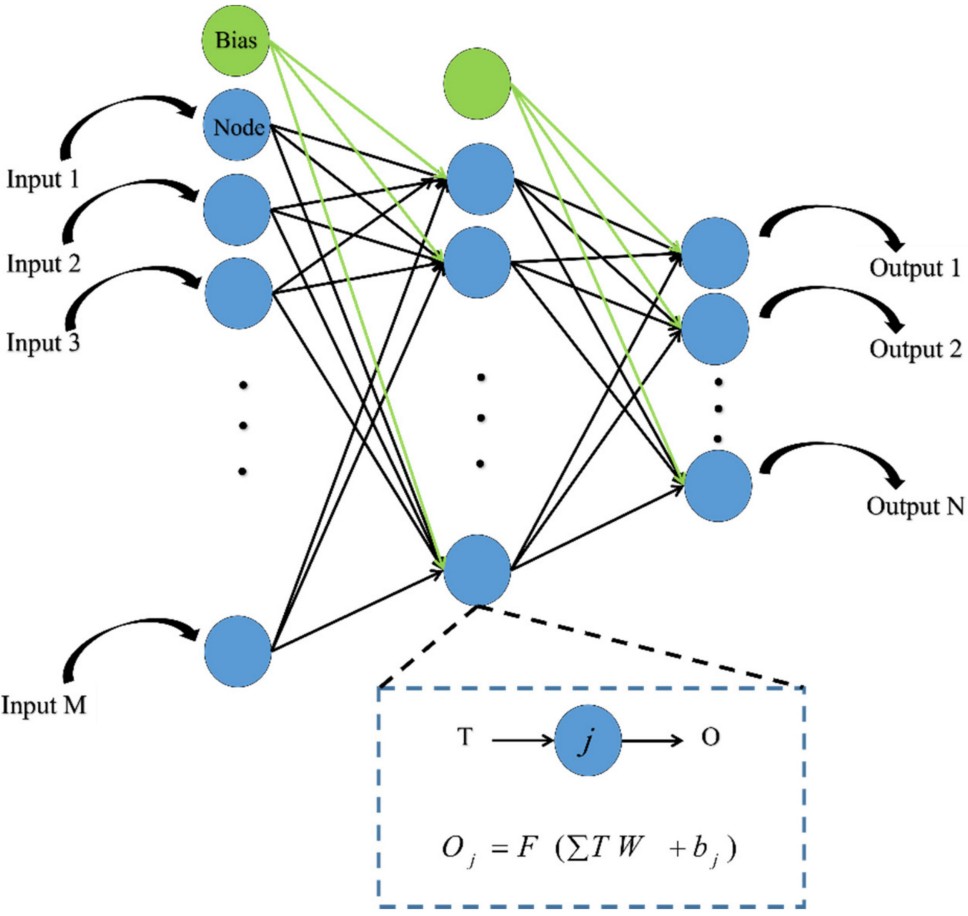

**Figure 2.** The structure of a single hidden layer MLP.

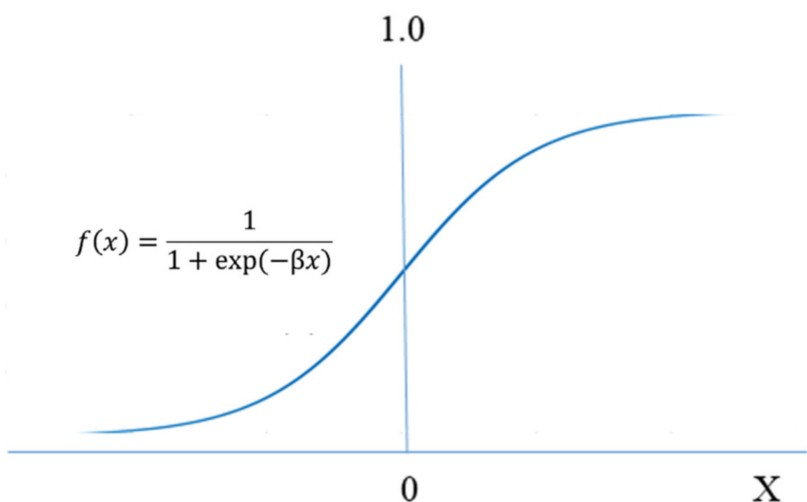

**Figure 3.** The Sigmoid function.

### 2.2.2. Metaheuristic Optimizers

This section briefly describes the metaheuristic techniques used. Generally speaking, the BBO, SSA, MFO, WDO are swarm-based approaches which mimic a specific nature-inspired process to implement the optimization. The word "optimization" here means finding the most optimal solution for the defined problem. Once they are combined with tools like SVMs [87] and ANNs [88], the objective is fine-tuning the influential parameters.

This measure enables the model to stay away from computational risks like local minima and dimension danger [89,90]. This process is detailed in the next section.

Proposed by Simon [91], the idea of the biogeography-based optimization is biogeography knowledge (i.e., which deals with the geographical dispersion of biological species. The individuals in the BBO are known as habitats. The conditions of residence for each zone and the habitability of these habitats are evaluated by two indices assigned to the geographic areas, namely habitat suitability index (HSI) and suitability index variable (SIV), respectively. Important decisions in this scheme are made based on the values of these indices by migration and mutation operators.

Salps are free-floating tunicates members of Salpidae family. Their foraging behavior is the basis of the SSA algorithm [92]. For better locomotion, they usually navigate in the form of so-called groups "salp chains". In the SSA, each individual indicates a candidate solution where they can be classified into two groups, namely followers and the leader. The leader is the first member of the chain and guides the followers toward food sources. Remarkably, due to the functioning of stochastic operators, the SSA can properly avoid local solutions that exist within multi-modal spaces [93].

Moths are known as fancy insects with high diversity in nature. The moth-flame optimization was presented by Mirjalili [94] based on the navigation scheme of moths. This scheme is named transverse orientation (TO). In this technique, the insects fly at night and use moonlight to travel long tracks by flying in a straight line. Considering the inefficiency of the TO, the moths are also lured by artificial light sources and try to fly in a straight direction through keeping a similar angle with the light. In the MFO, the individuals' positions represent the variables of the aimed problem.

The wind driven optimization is a heuristic population-based algorithm that is inspired by the atmospheric motions of air parcels with respect to the rotation of the globe. This algorithm was first suggested by Bayraktar, et al. [95] for electromagnetics usage. This method is designed as a combination of ideal gas state equation and Newton's second law. It is assumed that four forces in nature including pressure gradient force, frictional force, gravitational force, and Coriolis force are applied to the air parcels. More details, as well as the mathematical methodology of the mentioned algorithms, can be found in previous literature: BBO [96–98], SSA [99–101], MFO [102–104], and WDO [105–107].

### 2.2.3. Hybridization Process

As explained, the BBO, SSA, MFO, WDO metaheuristic algorithms are coupled with an ANN to create the BBO-MLP, SSA-MLP, MFO-MLP, and WDO-MLP ensembles. This task requires the MLP general structure to be yielded as the problem equation. Above all, based on a trial and error process, the most suitable complexity of the MLP (i.e., the number of hidden processors) was found to be 5. Therefore, the algorithms were applied to adjust 46 computational parameters (including ($7 \times 5 =$) 35 weights connecting the input and hidden neurons, ($5 \times 1 =$) 5 weights connecting the hidden and output neurons, 5 biases belonging to hidden layer, plus 1 bias in the output layer) to remedy the shortcomings of the neural approach.

Next, the same procedure was carried out for the metaheuristic ensembles to determine the optimal size of their population. Nine population sizes ($N_p$) in the range (10, 500) were tested for each ensemble. Similar to previous studies, they performed 1000 iterations to minimize the error. The RMSE was set as the objective function to calculate the learning error. The final RMSEs obtained for each algorithm, as well as the computation time, are shown in Table 1. According to this table, the best complexity of the BBO-MLP, SSA-MLP, MFO-MLP, and WDO-MLP is indicated by the $N_p$s of 300, 100, 100, and 100, respectively. Furthermore, the convergence (i.e., optimization) curves for the elite configurations are shown in Figure 4.

**Table 1.** The results of the sensitivity analysis.

| $N_P$ | BBO-MLP | | SSA-MLP | | MFO-MLP | | WDO-MLP | |
|---|---|---|---|---|---|---|---|---|
| | **RMSE** | **Time (s)** | **RMSE** | **Time (s)** | **RMSE** | **Time (s)** | **RMSE** | **Time (s)** |
| 10 | 5.505577 | 139.4756 | 6.355048 | 144.6717 | 5.8897 | 127.0382 | 6.215144 | 127.2421 |
| 25 | 4.730596 | 347.3891 | 5.748444 | 344.6891 | 5.540889 | 334.6007 | 5.942362 | 324.4255 |
| 50 | 4.974088 | 690.5993 | 5.569974 | 666.6998 | 5.610954 | 673.13 | 6.144947 | 632.0315 |
| 75 | 4.858011 | 1031.498 | 5.522838 | 953.5088 | 5.817214 | 1000.989 | 6.133931 | 986.733 |
| 100 | 4.816898 | 1310.805 | 4.920222 | 1263.715 | 5.530195 | 1255.081 | 5.664862 | 1398.88 |
| 200 | 4.74319 | 3025.854 | 5.10271 | 2532.209 | 5.815933 | 2505.335 | 6.285705 | 3270.84 |
| 300 | 4.692921 | 3870.514 | 5.104293 | 3770.992 | 5.728259 | 4225.847 | 6.166536 | 30,623.5 |
| 400 | 5.043938 | 5170.186 | 5.553021 | 5051.865 | 5.658828 | 5861.167 | 6.11727 | 5550.942 |
| 500 | 5.061255 | 6856.051 | 4.950267 | 7814.967 | 5.676524 | 6622.55 | 6.178576 | 6720.605 |

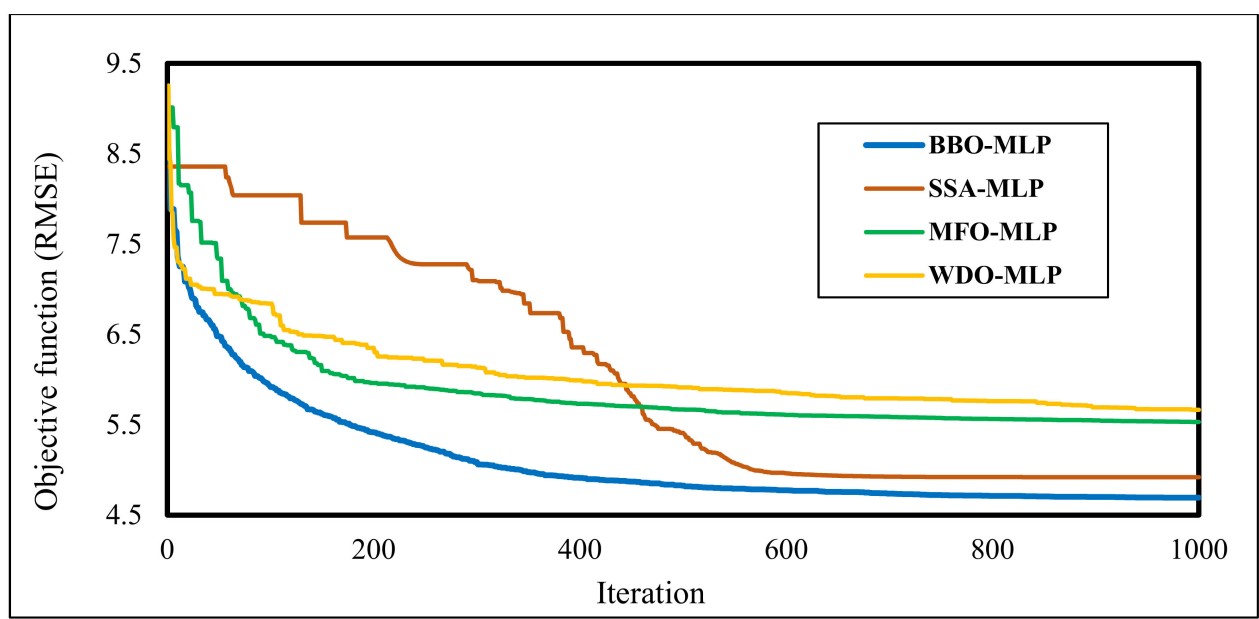

**Figure 4.** The convergence curves of the elite models.

## 3. Results and Discussion

### 3.1. Accuracy Criteria

Comparing the forecasted slumps with real ones is the principle of performance evaluation for the models. Two error criteria of *MAE* and *RMSE* are used for measuring the mean absolute and root mean square error values. Equations (1) and (2) express the formulation of the *RMSE* and *MAE*. Moreover, the coefficient of determination ($R^2$) is used to represent the correlation between the real slumps with forecasted values (see Equation (3)).

$$RMSE = \sqrt{\frac{1}{K}\sum_{i=1}^{K}\left[(Z_{i_{observed}} - Z_{i_{predicted}})\right]^2} \tag{1}$$

$$MAE = \frac{1}{K}\sum_{I=1}^{K}\left|Z_{i_{observed}} - Z_{i_{predicted}}\right| \tag{2}$$

$$R^2 = 1 - \frac{\sum\limits_{i=1}^{K} \left(Z_{i_{predicted}} - Z_{i_{observed}}\right)^2}{\sum\limits_{i=1}^{K} \left(Z_{i_{observed}} - \overline{Z}_{observed}\right)^2} \tag{3}$$

In the above equations, the number of instances is represented by $K$, the real and forecasted slump values are shown by $Z_{i\,predicted}$ and $Z_{i\,observed}$, respectively. In addition, $\overline{Z}_{observed}$ symbolizes the mean of real slump values.

### 3.2. Performance Evaluation

The prediction accuracy of the BBO-MLP, SSA-MLP, MFO-MLP, and WDO-MLP ensemble models was evaluated by comparing real slump values with their products. This process was executed for both training and testing phases to assess the ability of the models in inferring and generalizing the slump pattern. The results of the training data are shown in Figure 5 depicting a graphical comparison between the mentioned slump patterns.

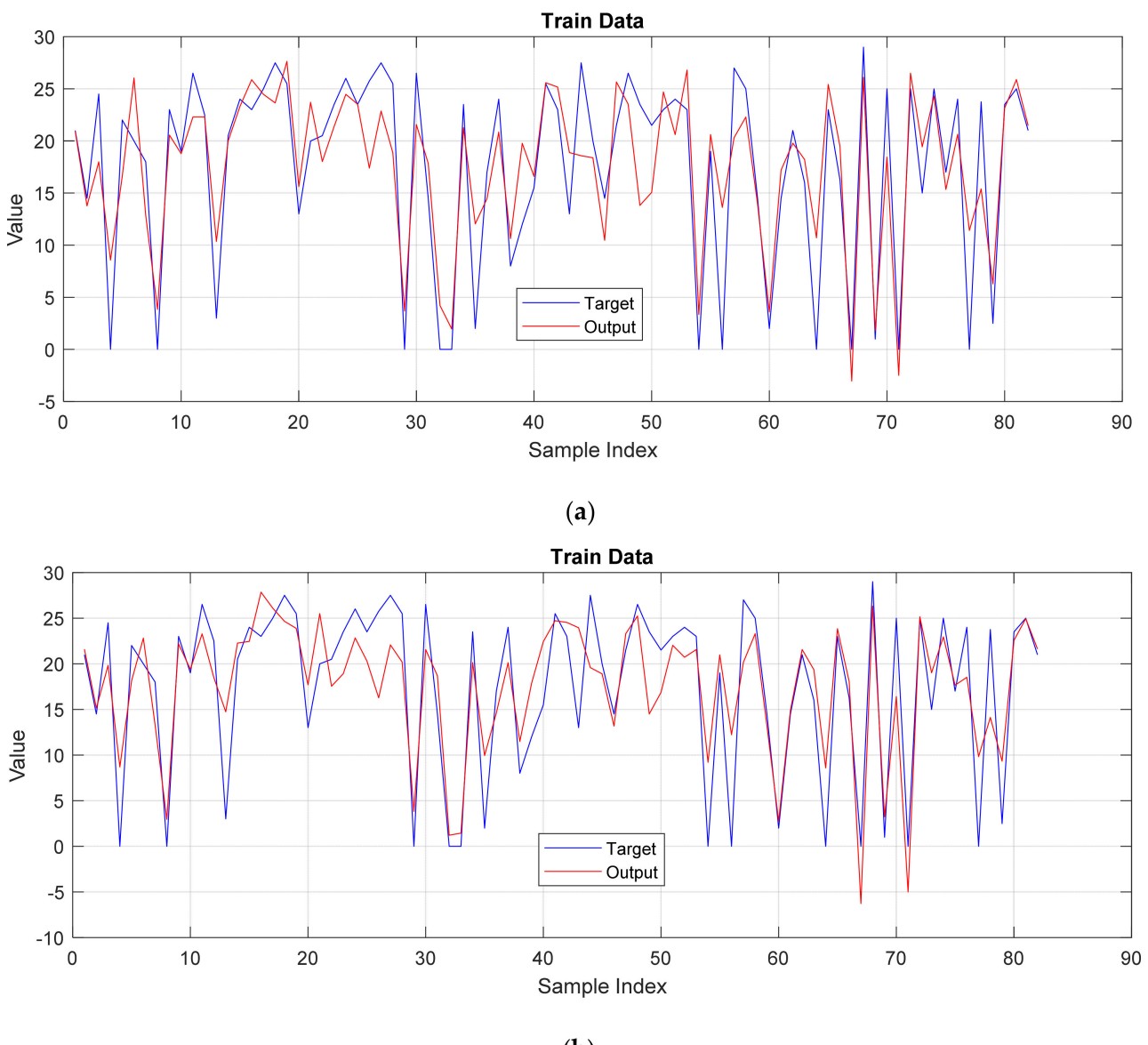

(**a**)

(**b**)

**Figure 5.** *Cont.*

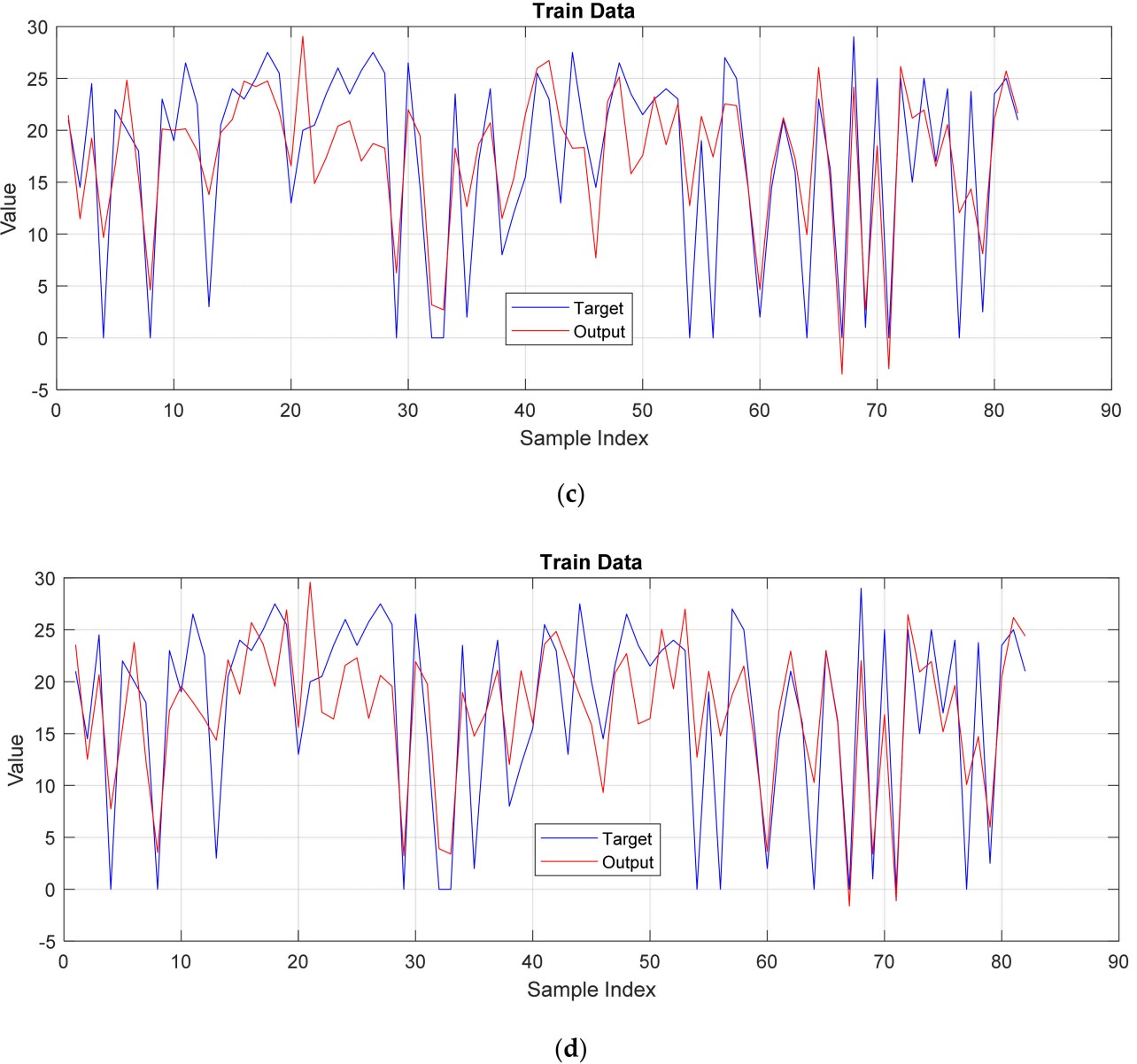

**Figure 5.** The training results obtained for (**a**) BBO-MLP, (**b**) SSA-MLP, (**c**) MFO-MLP, and (**d**) WDO-MLP predictions.

In this phase, obtained values of RMSE (4.6929, 4.9202, 5.5302, and 5.6649) indicate that all four models grasped a good perception of the slump–ingredients relationship. The calculated MAEs (3.6729, 3.8692, 4.3970, and 4.6087) are also evidence for a high level of accuracy. Regarding the correlation criterion, the obtained $R^2$ values (0.7479, 0.7202, 0.6472, and 0.6283) show more than 62% correlation for all predictive models.

The testing results also indicate that the models can successfully predict the slump of unfamiliar concrete specimens. The histogram chart showing the frequency of errors for each sample (=real slump − predicted slump) is presented in Figure 6. From error analysis, the error values range in (−4.255009938, 8.75706062), (−4.81094212, 10.4314921), (−4.494014358, 11.13801659), and (−5.211184587, 11.93617373).

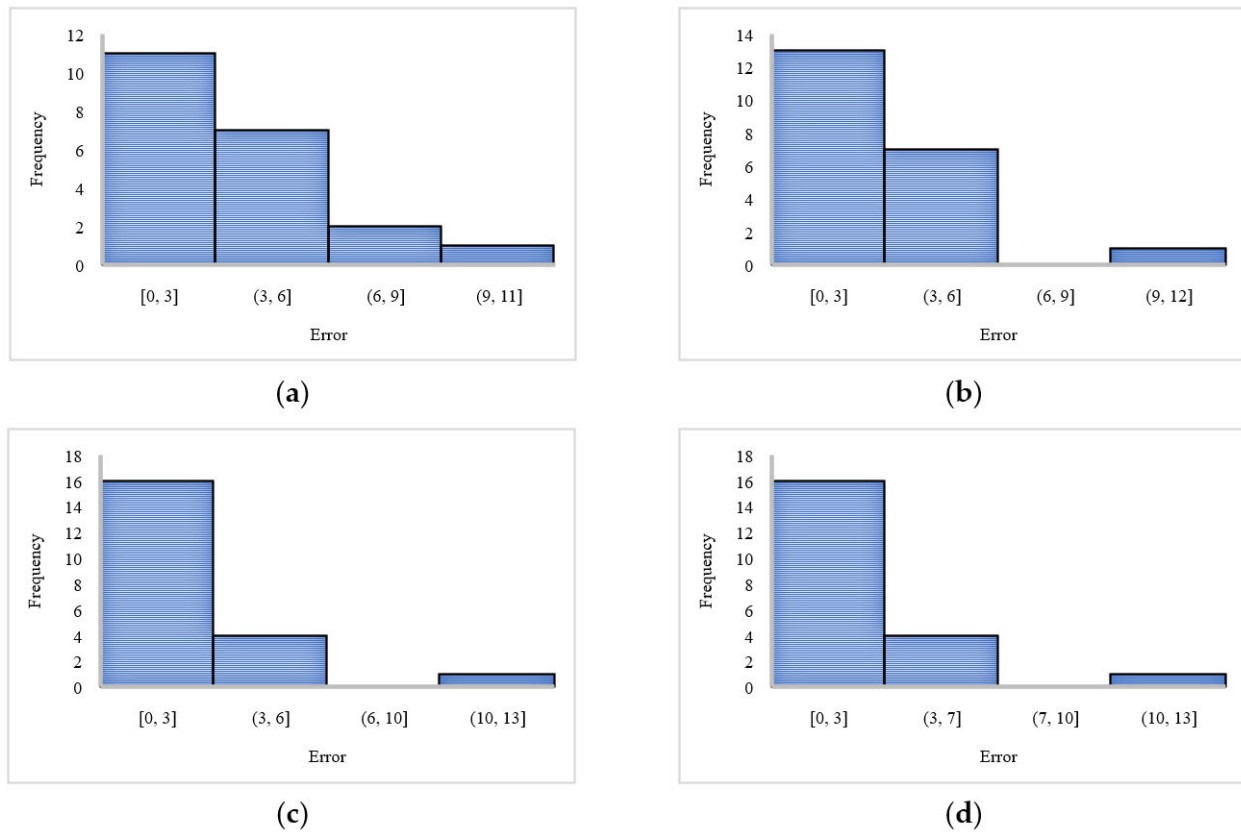

**Figure 6.** The testing errors histograms obtained for (**a**) BBO-MLP, (**b**) SSA-MLP, (**c**) MFO-MLP, and (**d**) WDO-MLP predictions.

Moreover, both error criteria of RMSE (3.6399, 3.8572, 3.3309, and 3.7540) and MAE (2.9521, 3.0871, 2.3156, and 2.8368) demonstrate that the models have presented a relatively accurate prediction of the slump. Furthermore, Figure 7 illustrates the correlation between the real and forecasted values in this phase. The $R^2$ values are 0.7157, 0.5793, 0.6748, and 0.6438.

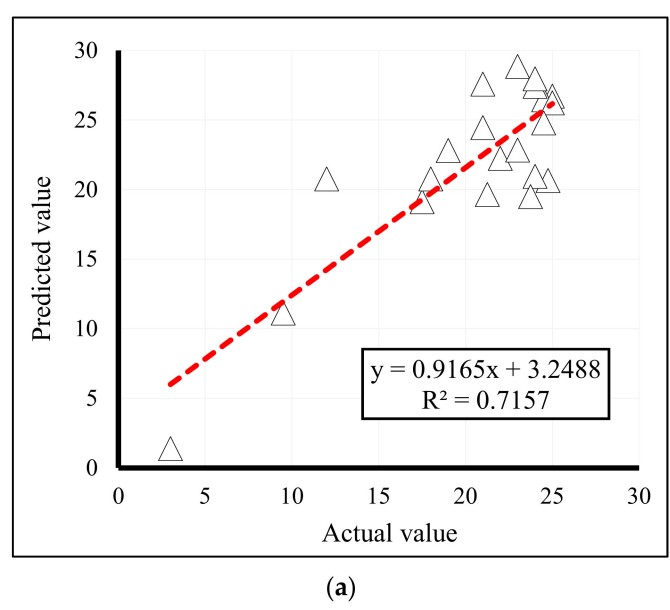

(**a**)

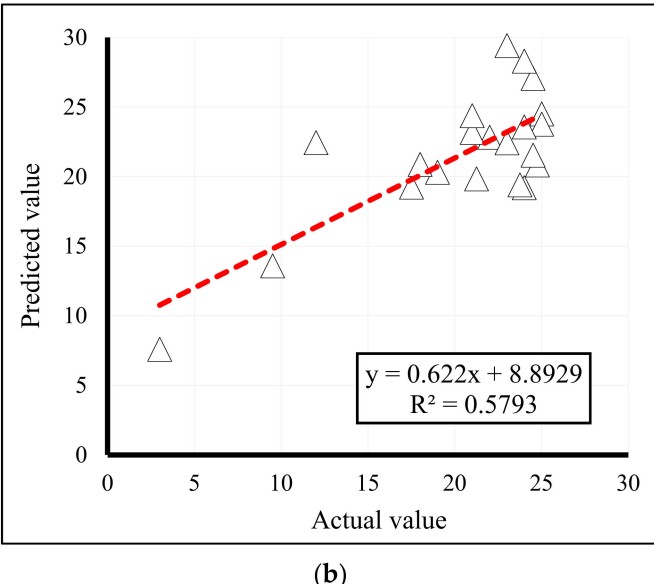

(**b**)

**Figure 7.** *Cont.*

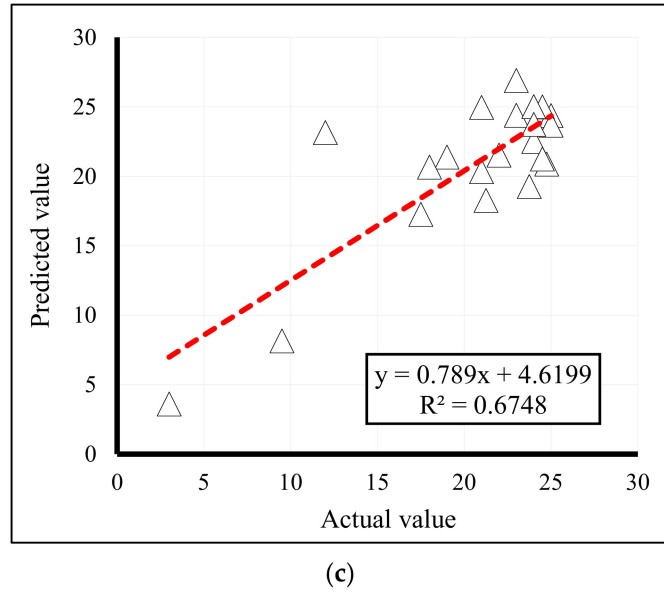

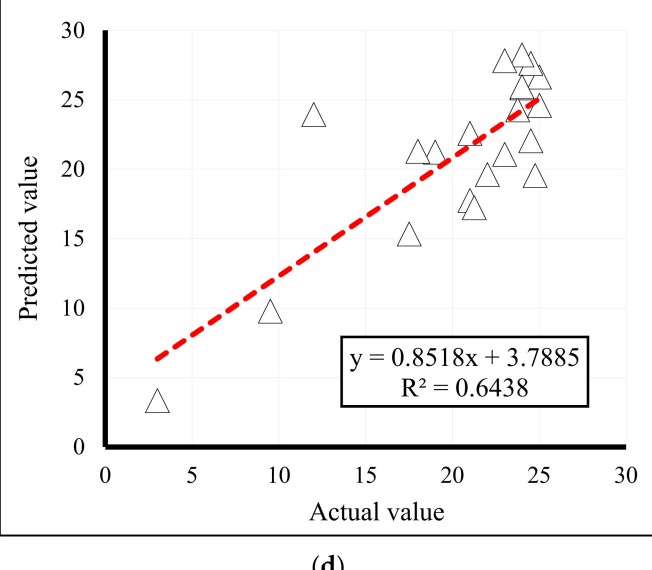

(**c**)                                           (**d**)

**Figure 7.** The correlation of testing samples for (**a**) BBO-MLP, (**b**) SSA-MLP, (**c**) MFO-MLP, and (**d**) WDO-MLP models.

Evaluation of the results (Table 2) shows that according to the MAE and RMSE criteria, the models have performed more efficiently in the generalizing the slump pattern. In other words, the error of testing samples is considerably lower than the training ones. However, the $R^2$s of the BBO-MLP and SSA-MLP indicate a lower correlation in the testing phase.

**Table 2.** The achieved accuracy results.

| Ensemble Models | Network Results | | | | | |
|---|---|---|---|---|---|---|
| | Training Phase | | | Testing Phase | | |
| | *RMSE* | *MAE* | $R^2$ | *RMSE* | *MAE* | $R^2$ |
| BBO-MLP | 4.6929 | 3.6729 | 0.7479 | 3.6399 | 2.9521 | 0.7157 |
| SSA-MLP | 4.9202 | 3.8692 | 0.7202 | 3.8572 | 3.0871 | 0.5793 |
| MFO-MLP | 5.5302 | 4.3970 | 0.6472 | 3.3309 | 2.3156 | 0.6748 |
| WDO-MLP | 5.6649 | 4.6087 | 0.6283 | 3.7540 | 2.8368 | 0.6438 |

Moreover, a well-known comparison model was hired to rank the used models and address the most reliable predictors. In this regard, the models were ranked based on the obtained accuracy criteria, and subsequently, three scores were assigned to each model in both phases. The results are shown in Figure 8. The calculated ranking scores (RSs) show the superiority of the BBO in training the ANN. After that, the SSA and MFO emerged as the second and third capable metaheuristic techniques. The performance of the WDO was weaker than three other colleagues in analyzing the intended parameters.

Focusing on testing scores, it can be seen that there are discrepancies between the performances of the models in two phases. Figure 8 outlines that the MFO-MLP (RS = 11) has the highest prediction capability. Considering both training and testing RSs, it can be determined that the BBO (RS = 21) constructs the most accurate MLP neural network, followed by the MFO (RS = 17), SSA (RS = 12), and WDO (RS = 10).

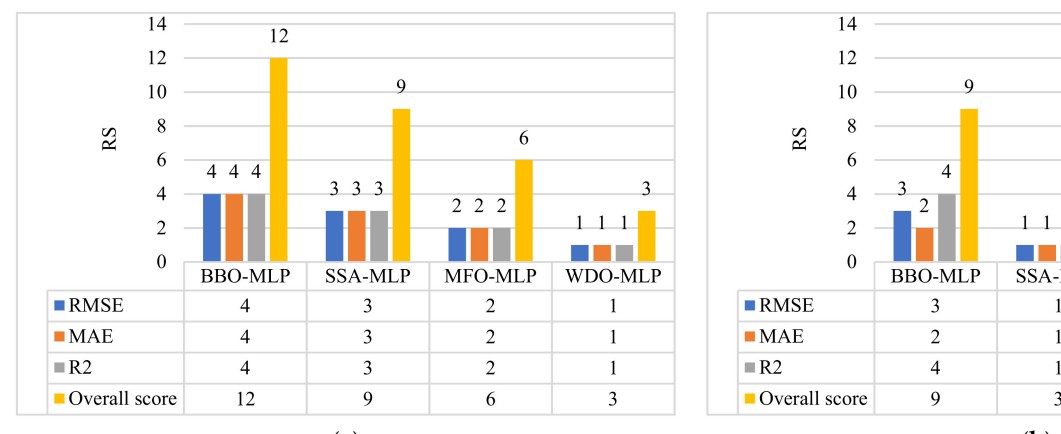 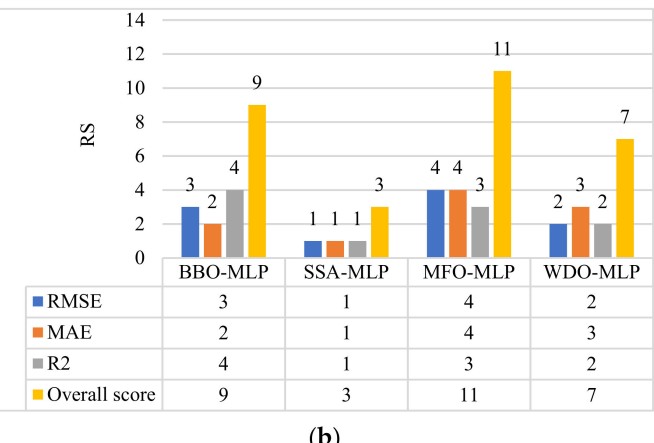

(**a**)  (**b**)

**Figure 8.** Executed ranking system based on RMSE, MAE, and $R^2$. (**a**) Training, and (**b**) Testing.

## 4. Conclusions

Slump is among the most important mechanical characteristics of concrete. In this effort, a reliable approximation of concrete slump was presented by considering the effect of seven concrete factors. Four capable ensembles, namely BBO-MLP, SSA-MLP, MFO-MLP, and WDO-MLP were successfully applied to this problem. In the computational sense, optimizing the hyper parameters showed that the population size required by the optimal BBO is 3 times larger than other algorithms (300 versus 100). On the other hand, prediction results revealed that the combination of machine learning and metaheuristic techniques obtains a reliable understanding of the dependency of the slump on the concrete ingredients. The MLPs constructed by the BBO and MFO were the most efficient models in the training and testing phases, respectively. Hence, the authors would recommend these ensembles as inexpensive yet accurate tools for achieving an optimum design of concrete mixture through estimating the concrete slump. Lastly, there are some ideas that are worth investigating in future studies. Taking external datasets for increasing the generalizability, feature selection for simplifying the problem space, and comparative efforts for discovering the strongest algorithms are strongly suggested.

**Author Contributions:** H.S.: methodology, writing—original draft preparation, software, resources. F.N.: investigation, data curation, methodology, validation. M.L.N.: supervision, project administration, funding acquisition, writing of final manuscript. All authors have read and agreed to the published version of the manuscript.

**Funding:** This research received no external funding.

**Institutional Review Board Statement:** Not applicable.

**Informed Consent Statement:** Not applicable.

**Data Availability Statement:** The data that support the findings of this study are available from the corresponding author upon request.

**Conflicts of Interest:** The authors declare no conflict of interest.

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
