# Peer review of "Indirect Analysis of Concrete Slump Using Different Metaheuristic-Empowered Neural Processors"

_sustainability, doi:10.3390/su141610373_

Round 1

Reviewer 1 Report

The article titled “Indirect Analysis of Concrete Slump Using Different Metaheuristic-Empowered Neural Processor” works on the neural network computing of slump of fresh concrete considering a wide range of parameters such as the influence of cement, water, slag, fly ash, fine aggregate, superplasticizer, and coarse aggregate and their different ratios. The manuscript is written in clear language, and no issues with grammar were found. The introduction section briefly provides research background, and literature on neural computing techniques in concrete technology, and leads to a clear definition of goals and scope. The research is interesting, novel, and has practical implications. Thus, I recommend the publication of this article after minor changes.

1)     In the introduction remove expressions like “Scholars like”. Instead explain the each source with clear details.

2)     Page 2: Moayedi, Kalantar, Foong, Tien Bui and Motevalli, not a correct representation of citation in a scientific article. Same comment for entire article.

3)     Figure 1; provide high quality and sharp files.

4)     Reduce the size of equations 1,2 and 3 to match closely with main text.

5)     Conclusion section is poorly written.

6)     More studies in literature are needed from the past two years (2020s).

Author Response

The article titled “Indirect Analysis of Concrete Slump Using Different Metaheuristic-Empowered Neural Processor” works on the neural network computing of slump of fresh concrete considering a wide range of parameters such as the influence of cement, water, slag, fly ash, fine aggregate, superplasticizer, and coarse aggregate and their different ratios. The manuscript is written in clear language, and no issues with grammar were found. The introduction section briefly provides research background, and literature on neural computing techniques in concrete technology, and leads to a clear definition of goals and scope. The research is interesting, novel, and has practical implications. Thus, I recommend the publication of this article after minor changes.

1)     In the introduction remove expressions like “Scholars like”. Instead explain the each source with clear details.

Answer: Thank you for this valuable comment. The requested changes are applied throughout the text. Please find some corrections below:

“More recently, highly non-linear and complicated engineering problems have driven scientists to employ metaheuristic techniques for diverse optimization objec-tives [57-61]. Focusing on civil engineering domain, their application covers a broad range from structural health monitoring [62] to material strength modeling [63]. In these algorithms, optimal solution is sought by a set of population that update their situation within the defined search space. Thus, they can properly assist intelligent models to attain a system preserved from prevalent training issues [64-66].

In the case of concrete parameters modeling, a metaheuristic algorithm tries to establish the best contribution between the mixture gradients and the purposed pa-rameter. For instance, Bui, et al. [67] applied whale optimization algorithm to an ANN for predicting CS. This algorithm achieved root mean square error (RMSE) of 2.6985 which was comparably lower than the RMSE of dragonfly algorithm (3.3325) and ant colony optimization (3.4452). In another work by Bui, et al. [68], modified firefly algo-rithm (MFA) was assigned to the similar task for high performance concrete. They compared this model with those in several precious studies (e.g., genetic program-ming) and found that the MFA-ANN is superior to them. Zhao, et al. [69] proved the proficiency of equilibrium optimizer for analyzing the tensile strength. The hybrid model could improve the performance of ANN (e.g., correlation rose from 0.89 to 0.92).”

Kindly, as you certainly know, in some cases it is inevitable to cite some references altogether as they all imply the same concept related to the sentence.

2)     Page 2: Moayedi, Kalantar, Foong, Tien Bui and Motevalli, not a correct representation of citation in a scientific article. Same comment for entire article.

Answer: Thank you for this valuable comment. The reference style is corrected to avoid such issues all over the manuscript.

3)     Figure 1; provide high quality and sharp files.

Answer: Thank you for this valuable comment. Figures 5,3,2 are replaced with more vivid versions. Other are original versions.

4)     Reduce the size of equations 1,2 and 3 to match closely with main text.

Answer: Thank you for this valuable comment. Corrected as requested.

5)     Conclusion section is poorly written.

Answer: Thank you for this valuable comment. Conclusion is rewritten to be more informative.

“Slump is among the most important mechanical characteristics of concrete. In this effort, a reliable approximation of concrete slump was presented by considering the effect of seven concrete factors. Four capable ensembles, namely BBO-MLP, SSA-MLP, MFO-MLP, and WDO-MLP were successfully applied to this problem. In the computational sense, optimizing the hyperparameters showed that the population size required by the optimal BBO is 3 times larger than other algorithms (300 versus 100). On the other hand, prediction results revealed that the combination of machine learning and metaheuristic techniques acquires a reliable understanding from the dependency of slump on the concrete ingredients. The MLPs constructed by the BBO and MFO were the most efficient models in the training and testing phases, respectively. Hence, the authors would recommend these ensembles as inexpensive yet accurate tools for achieving an optimum design of concrete mixture through estimating the concrete slump. Lastly, there are some ideas that are worth investing in future studies. Taking external datasets for increasing the generalizability, feature selection for simplifying the problem space, and dedicating comparative efforts for discovering the strongest algorithms are highly suggested.”

6)     More studies in literature are needed from the past two years (2020s).

Answer: Thank you for this valuable comment. We comprehensively modified literature review by adding newer works from various domains including background of recent technology, artificial intelligence, etc. Some of them most related to concrete works are presented below:

  1. Predicting the slump of industrially produced concrete using machine learning: A multiclass classification approach
  2. Machine learning to predict properties of fresh and hardened alkali-activated concrete
  3. Compressive strength prediction of lightweight concrete: machine learning models
  4. Data-driven model for ternary-blend concrete compressive strength prediction using machine learning approach

Lastly, the authors would like to sincerely appreciate your kind review on our work which led to improving the quality. We hope the revised version is acceptable in your opinion.

Reviewer 2 Report

Dear Authors,

The article submitted for review entitled: Indirect Analysis of Concrete Slump Using Different Metaheuristic Empowered Neural Processor presents an interesting approach to the topic presented.

The paper consists of 3 main chapters, which are structured into a logical whole. The strengths of the study are the neural network models that describe the presented issue.

Despite this, the article has several shortcomings that should be taken into account in order to improve its attractiveness to the reader:

- There are places in the paper where there are too many spaces between words.

- Spaces should be inserted between tinings placed in square brackets - it reads better.

- Mathematical equations should be centered and should be corrected for their appearance, they seem to be stretched.

- A space should be inserted between the description of variables occurring in mathematical relationships and the next paragraph. This form will allow the reader to separate the text of the article from the description of the mathematical relationships.

- Figure captions should not end with periods - e.g.: figure 1

- Captions over tables should not end with periods - e.g.: table 1

- Literature should be corrected according to the publisher's recommendations

Summary: 

In order to continue the publication process further, the reviewer would like to encourage the authors to follow the comments/suggestions. A thorough revision of the paper is required. 

Author Response

Dear Authors,

The article submitted for review entitled: Indirect Analysis of Concrete Slump Using Different Metaheuristic Empowered Neural Processor presents an interesting approach to the topic presented.

The paper consists of 3 main chapters, which are structured into a logical whole. The strengths of the study are the neural network models that describe the presented issue.

Despite this, the article has several shortcomings that should be taken into account in order to improve its attractiveness to the reader:

- There are places in the paper where there are too many spaces between words.

Answer: Thank you for this valuable comment. We double-checked the structure of the paper and made some corrections. Kindly, we believe some issues are related to the template and can be corrected during proofreading.

- Spaces should be inserted between tinings placed in square brackets - it reads better.

Answer: Thank you for this valuable comment. Kindly, we are not sure if we have understood your point rightly. But we checked all brackets. Those related to references are organized by the Endnote software automatically and those like [137.0, 374.0] have a space between two values showing the range of data.

- Mathematical equations should be centered and should be corrected for their appearance, they seem to be stretched.

Answer: Thank you for this valuable comment. Equations 1, 2, 3 are resized to properly fit the paper dimensions and also centered.

- A space should be inserted between the description of variables occurring in mathematical relationships and the next paragraph. This form will allow the reader to separate the text of the article from the description of the mathematical relationships.

Answer: Thank you for this valuable comment. It is a good suggestion. Applied as requested.

- Figure captions should not end with periods - e.g.: figure 1

Answer: Thank you for this valuable comment. Corrected for all figures.

- Captions over tables should not end with periods - e.g.: table 1

Answer: Thank you for this valuable comment. Corrected for all tables.

- Literature should be corrected according to the publisher's recommendations

Answer: Thank you for this valuable comment. The specific style of the journal is applied for all. Kindly, more corrections can be done during proofreading process.

Summary: 

In order to continue the publication process further, the reviewer would like to encourage the authors to follow the comments/suggestions. A thorough revision of the paper is required.

Lastly, the authors would like to sincerely appreciate your kind review on our work which led to improving the quality. We hope the revised version is acceptable in your opinion.